# Identification of Shemin pathway genes for tetrapyrrole biosynthesis in bacteriophage sequences from aquatic environments

Helen Wegner [1,7], Sheila Roitman [2,6,7], Anne Kupczok [3], Vanessa Braun[1], Jason Nicholas Woodhouse[4], Hans-Peter Grossart [4,5], Susanne Zehner [1], Oded Béjà [2] & Nicole Frankenberg-Dinkel [1] ✉

Tetrapyrroles such as heme, chlorophyll, and vitamin $B_{12}$ are essential for various metabolic pathways. They derive from 5-aminolevulinic acid (5-ALA), which can be synthesized by a single enzyme (5-ALA synthase or AlaS, Shemin pathway) or by a two-enzyme pathway. The genomes of some bacteriophages from aquatic environments carry various tetrapyrrole biosynthesis genes. Here, we analyze available metagenomic datasets and identify *alaS* homologs (viral *alaS*, or v*alaS*) in sequences corresponding to marine and freshwater phages. The genes are found individually or as part of complete or truncated three-gene loci encoding heme-catabolizing enzymes. Amino-acid sequence alignments and three-dimensional structure prediction support that the v*alaS* sequences likely encode functional enzymes. Indeed, we demonstrate that is the case for a freshwater phage v*alaS* sequence, as it can complement an *Escherichia coli* 5-ALA auxotroph, and an *E. coli* strain overexpressing the gene converts the typical AlaS substrates glycine and succinyl-CoA into 5-ALA. Thus, our work identifies v*alaS* as an auxiliary metabolic gene in phage sequences from aquatic environments, further supporting the importance of tetrapyrrole metabolism in bacteriophage biology.

Tetrapyrroles are natural products considered as 'pigments of life', as they contribute to the functionality of essential biological processes such as respiration, photosynthesis, and methanogenesis[1,2]. Tetrapyrroles are also involved in oxygen transport, catalysis, light-sensing, and -harvesting, to name just a few. All tetrapyrroles, including chlorophyll, heme, siroheme, corrin (including vitamin $B_{12}$), coenzyme $F_{430}$, heme *d1*, and bilins, derive from a common precursor, i.e. 5-aminolevulinic acid (5-ALA). Two known routes lead to the formation of 5-ALA, the Shemin (or C4) pathway and the C5 pathway[3,4]. The Shemin pathway uses a single enzyme, 5-ALA synthase (AlaS), to convert the two substrates glycine and succinyl-Coenzyme A (succinyl-

CoA) to 5-ALA in a reaction dependent on pyridoxal 5′-phosphate (PLP) (Fig. 1). This pathway is utilized by alphaproteobacteria and most eukaryotes, except plants. All other bacteria, archaea, and plants use the C5 pathway, in which two enzymes drive the conversion of the C5 carbon skeleton of glutamate to 5-ALA. The reaction starts with a rather unusual substrate, glutamyl-tRNA, which is transformed into glutamate semialdehyde (GSA) by an enzyme called glutamyl-tRNA reductase (GtrR). GSA is subsequently converted to 5-ALA by GSA aminomutase (GsaM). Starting off with 5-ALA, all modified tetrapyrroles are then synthesized along a branched biochemical pathway[2] (Fig. 1).

[1]Department of Biology, Microbiology, University of Kaiserslautern-Landau, Kaiserslautern, Germany. [2]Faculty of Biology, Technion-Israel Institute of Technology, Haifa, Israel. [3]Department of Plant Sciences, Bioinformatics, Wageningen University & Research, Wageningen, Netherlands. [4]Department of Plankton and Microbial Ecology, Leibniz Institute of Freshwater Ecology and Inland Fisheries, Stechlin, Germany. [5]Institute of Biochemistry and Biology, Potsdam University, Potsdam, Germany. [6]Present address: Department of Molecular Biology, Max Planck Institute for Biology, Tübingen, Germany. [7]These authors contributed equally: Helen Wegner, Sheila Roitman ✉e-mail: nicole.frankenberg@rptu.de

Almost two decades ago, it was noted that genomes of some viruses that infect cyanobacteria (cyanophages) possess tetrapyrrole biosynthesis genes[5–7]. These bacteriophages are the most abundant biological entities in aquatic environments[8,9]. They affect microbial populations by infecting and killing bacterial hosts, facilitating horizontal gene transfer, modifying host metabolism, and releasing bacterial-derived (organic) matter through host cell lysis[10,11]. The tetrapyrrole biosynthesis genes identified thus far in phages encode heme oxygenases (HemO) and ferredoxin-dependent bilin reductases (FDBRs) both involved in the biosynthesis of linear tetrapyrrole pigments, which are important for light-harvesting and -sensing. Most of these so-called auxiliary metabolic genes (AMGs) encode enzymes that mimic host enzyme activities, such as the FDBR PcyA, which is involved in the biosynthesis of the turquoise pigment phycocyanobilin (PCB) (Fig. 1)[5,7]. Some phage-encoded FDBR genes, however, are unique because they combine the enzymatic activity of two host enzymes in a single polypeptide. Thus far, these sequences (i.e., *pebS* and *pcyX*) are unique to phages and have not been found in bacteria. Both genes encode related enzymes (PebS and PcyX) catalyzing the biosynthesis of the light-harvesting chromophore phycoerythrobilin (PEB), a pink pigment native to cyanobacterial light-harvesting complexes[5,6] (Fig. 1).

Analysis of cyanophage, pelagiphage (phage infecting the abundant SAR11 bacterial lineage) and phage metagenomic sequences suggest the presence of additional tetrapyrrole pathway genes, including heme and vitamin $B_{12}$ biosynthesis[7,12,13]. Overall, AMGs are highly abundant and are thought to augment bacterial metabolism during infection[10,11]. In cyanophages, for instance, many AMGs encoding photosynthetic reaction centres and electron transfer proteins are thought to redirect electron flow during infection to enhance ATP production in cyanobacteria[11,14,15]. This will ensure sufficient energy supply for phage progeny production during the lytic infection cycle.

Here, we report the identification and characterization of viral encoded 5-ALA synthase (vAlaS) of the Shemin pathway from bacteriophages and its distribution within extensive metagenomic datasets. *valaS* sequences often occur in a three-gene cassette together with genes encoding a heme oxygenase and a PcyA or PcyX (PcyA/X)- like FDBR; however, stand-alone or broken cassettes are also observed. We show that vAlaS is a functional enzyme and is even able to complement the lack of the tRNA^Glu-dependent C5 pathway in bacteria. Our results indicate the importance of stable tetrapyrrole metabolism during infection, as phages acquired *valaS* via horizontal gene transfer and have maintained this gene. It also suggests that different bacteria may be infected by phages bearing *valaS*, regardless of the bacteria's specific 5-ALA synthesis pathway.

## Results

### Phage-encoded AlaS sequences are widely distributed in marine and freshwater environments

In the past, we described the identification of heme-derived pigment biosynthesis genes in viral-enriched metagenomic datasets. As a continuation of this research and during the work on an unrelated AMG[6,16,17], a reassembled *Tara* Oceans database and publicly available JGI GOLD datasets were screened for AMGs. In addition to the previously identified *hemO* and FDBR (*pcyX*) cassette, we recognized a third gene encoding a putative 5-aminolevulinic acid synthase (hereafter referred to as *valaS*), the key enzyme of the Shemin pathway of tetrapyrrole biosynthesis[3]. Using this vAlaS as bait, we expanded our search using the same procedure and retrieved several new contigs carrying the three heme metabolism-related genes either as a three-gene cassette (*hemO*, *pcyA/X*, *valaS*), a broken cassette (*hemO* and *pcyA/X* together, *valaS* nearby) or individual *valaS* from metagenomic datasets (Fig. 2; Supplementary Data 1). These contigs originate from the Arctic, Pacific, Indian, and Atlantic oceans, salt ponds, peat bogs, and freshwater lakes. The translated viral *hemO* and *pcyA/X* (v*hemO*,

v*pcyA/X* thereafter) sequences from marine samples resemble those found in the recently isolated Mosig pelagiphage[12], suggesting a pelagiphage origin (Supplementary Figs. S1, S2). The phylogeny of gp13, a phage capsid neck protein, found in phages of the *Myoviridae* phenotype, further confirmed this assumption, since all our retrieved contigs cluster together with the two isolated pelagiphages with *myoviridae* morphology (Supplementary Fig. S3). One of the contigs analyzed was from a previously reported manually curated phage genome CB_2 from a peatbog metagenome[13]. Furthermore, *valaS* sequences were also found in a second peatbog lake Grosse Fuchskuhle, as part of the LIMNOS German lake dataset (PRJEB47226; Supplementary Data 1 and 2). Phylogeny of AlaS protein sequences reveals that the bacteriophage-encoded enzymes cluster separately from the bacterial ones, except for an alphaproteobacteria *Rhodobacter*-like genomic bin, assembled from marine sponge symbionts (Fig. 3). To estimate the relative abundance of *valaS* sequences in the environment, we compared it to *psbA* (encoding the D1 protein of photosystem II), carried by cyanophages, probably the most abundant AMG. We found that in marine samples v*psbA* is 60 to 175,682 times more abundant than v*alaS*, with an average of 27,618 and a median of 15,541. In our freshwater samples, originating from lakes around Berlin, Germany, v*psbA* is 4.8 to 556 times more abundant than v*alaS*, with an average of 134 and a median of 117 (Supplementary Data 3).

### Phylogeny and 3-D structure prediction suggest that vAlaSs are functional enzymes

vAlaS protein sequences exhibited the highest amino acid sequence identity (~55%) to AlaS sequences from the *Pelagibacteraceae*, a family of the alphaproteobacteria. This family exclusively utilizes the Shemin pathway to synthesize 5-ALA. AlaS from the alphaproteobacterium *Rhodobacter capsulatus* is thus far the best characterized bacterial AlaS member and served as a control in our experiments[18–21]. It catalyzes the condensation of succinyl-CoA and glycine to 5-ALA, the common precursor of all tetrapyrroles. Bacterial AlaS (EC 2.3.1.37) has been studied in detail and was shown to be a pyridoxal 5′-phosphate (PLP)-dependent homodimeric enzyme consisting of two ~44 kDa monomers. The PLP cofactor is involved in binding the substrate glycine to form an aldimine to allow reaction with the second substrate succinyl-CoA. Overall, major catalytic residues, including the conserved lysine residue for PLP-cofactor binding as well as the substrate binding site, are highly conserved in the viral versions of the protein, implying that they are functional enzymes (Supplementary Fig. S4, Supplementary Data 5). This is further strengthened by an AlphaFold-generated structural model of vAlaS (CB_2) that superimposes perfectly the crystal structure of *R. capsulatus* AlaS (RcA) with a root mean square distance (RMSD) of 0.875 Å (Fig. 4; Supplementary Fig. S5, Supplementary Data 7). Interestingly, structure prediction and amino acid sequence alignment further revealed that many identified vAlaS sequences lack the major regulatory site for heme feedback inhibition, i.e., one or both homologs of the amino acid residues His342 and Cys400 (RcA sequence numbering; see also Supplementary Fig. S4). Therefore, vAlaS may not respond to heme feedback inhibition, which could be beneficial during infection, as a steady flow through the pathway will be necessary to maintain tetrapyrrole levels.

### vAlaS is an active enzyme and functionally complements a 5-ALA auxotrophic *E. coli* mutant

We performed functional complementation to assess vAlaS activity using a 5-ALA-auxotrophic *E. coli* strain (*E. coli* ST18) lacking the *gtrR* gene of the C5 pathway[22]. This mutant is only viable if grown with 5-ALA supplementation in the medium or when expressing a functional pathway for the synthesis of 5-ALA. Although *E. coli* employs the C5 pathway, a functional AlaS from the Shemin pathway can complement the 5-ALA-auxotrophic phenotype[23,24] (Supplementary Fig. S4a). The plasmid encoded vAlaS under *tac* promoter control was able to rescue

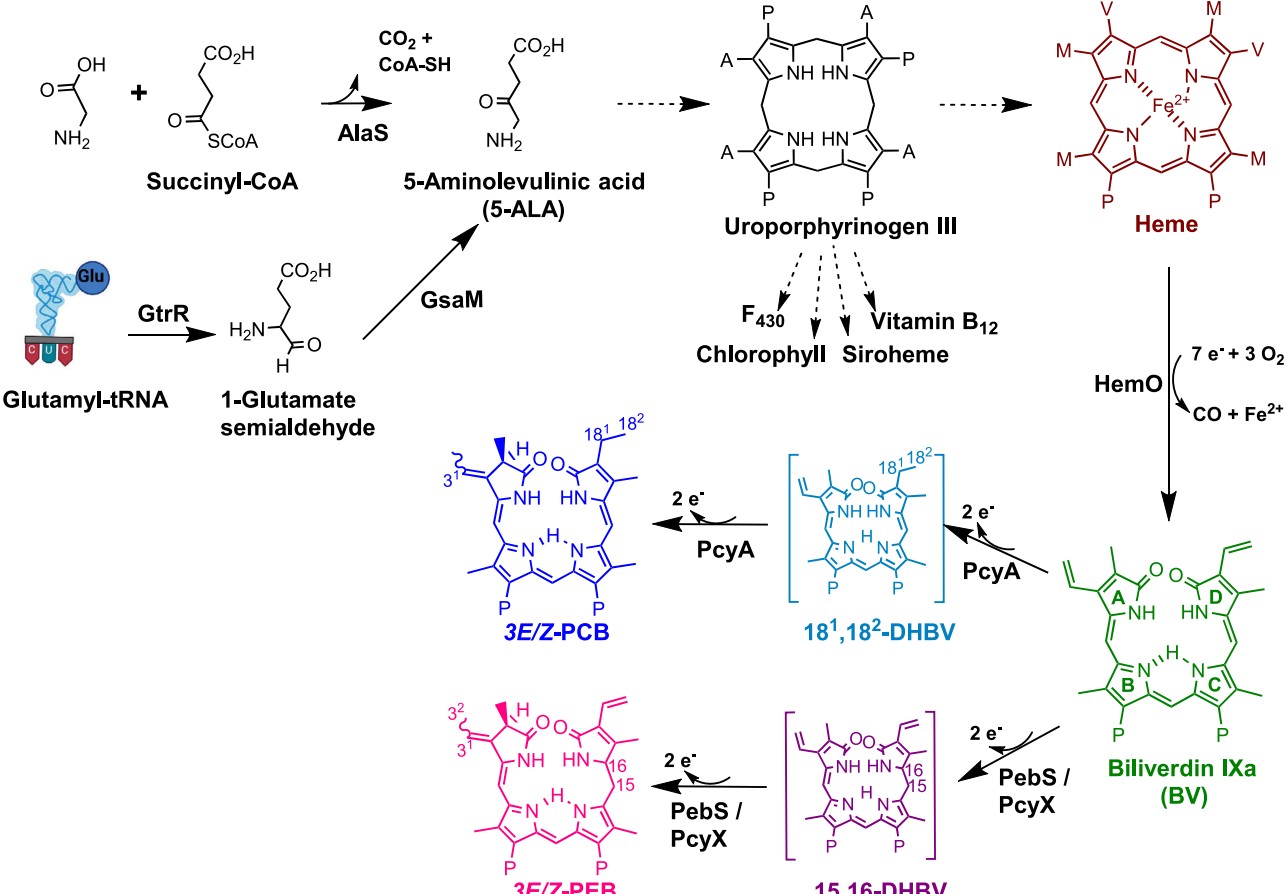

**Fig. 1 | Tetrapyrrole biosynthesis pathway.** The common precursor 5-aminolevulinic acid (5-ALA) is synthesized by two different pathways. While plants, most bacteria and archaea use the C5 pathway employing glutamyl t-RNA reductase (GtrR) and glutamate-1-semialdehyde-2,1-aminomutase (GsaM), the aminolevulinic acid synthase (AlaS) within the Shemin pathway of alphaproteobacteria, mammals and birds condenses glycine and succinyl-CoA to yield 5-ALA. Eight molecules of 5-ALA are needed to synthesize intermediate uroporphyrinogen III, an intermediate to most modified tetrapyrroles. Heme can then be converted

through heme oxygenase (HemO) to linear biliverdin IXα, which can be converted through ferredoxin-dependent bilin reductases to phycocyanobilin (PCB) or phycoerythrobilin (PEB). PcyA=phycocyanobilin:ferredoxin oxidoreductase; PebS/PcyX=phycoerythrobilin synthase; DHBV=dihydrobiliverdin. The side chains are abbreviated as follows: P=propionate; A=acetate; M=methyl; V=vinyl. Created with parts from BioRender.com, released under a Creative Commons Attribution-NonCommercial-NoDerivs 4.0 International license.

the ALA auxotrophic *E. coli* ST18 strain. The basal activity of the *tac* promoter was sufficient to enable functional complementation with a calculated growth rate being only slightly lower at $\mu = 0.56\,h^{-1}$ for the complemented strain compared to ST18 supplemented with 5-ALA ($\mu = 0.67\,h^{-1}$). The observed slightly slower growth of the complemented strain upon addition of inducer is probably due to increased metabolic burden overproducing the protein (Fig. 5a). In contrast, the active site variant vAlaS_K250A was not able to complement the ALA auxotrophy of *E. coli* ST18. This variant possesses a mutation in the essential lysine residue (K250) of the PLP cofactor binding site, resulting in an inactive enzyme. Both strains, the strain containing the empty vector and the inactive variant remained auxotrophic and only underwent 1-2 cell divisions, probably due to residual heme in the cells from the inoculum.

### The complemented *E. coli* 5-ALA auxotroph is able to synthesize 5-ALA from glycine and succinyl-CoA
Next, we explored whether we can detect vAlaS activity in bacterial lysate originating from the complementation culture. As *E. coli* is unable to use the AlaS substrates glycine and succinyl-CoA to synthesize 5-ALA, the observed formation of 5-ALA is due to the presence of vAlaS. The highest product formation (i.e., 5-ALA) was observed after 15 min, which steadily declined due to subsequent further

metabolism towards heme and other tetrapyrroles (Fig. 5b). Our results are further supported by the accumulation of porphyrins in the culture supernatant (reddish color) and the thus observed light sensitivity of the complemented cultures. Increased 5-ALA concentration leads to deregulation of the tetrapyrrole biosynthesis pathway, affecting the metabolic flux through the pathway and the accumulation of photoactive porphyrins[25] (Supplementary Fig. S6b). Therefore, all complementation experiments were protected from direct exposure to light. The accumulation of porphyrins could not be reduced by coexpression of a phage heme oxygenase and FDBR (Supplementary Fig. S7). Taken together, these results demonstrate that vAlaS is indeed a functional enzyme that converts glycine and succinyl-CoA to 5-ALA and thereby adds to the list of identified and enzymatically characterized viral AMGs.

### Freshwater phages possess three genes encoding functional tetrapyrrole biosynthesis enzymes
The *valaS* characterized here originated from metagenomic DNA attributed to the CB_2 phage[13], which in addition to *valaS* also possesses the two other heme metabolic genes, *vhemO* and *vpcyA/X*, as a broken cassette. Thus far, the derived proteins of the latter have only been characterized from marine environments[6]. The phylogeny of HemO sequences shows a less clear distinction between phage and

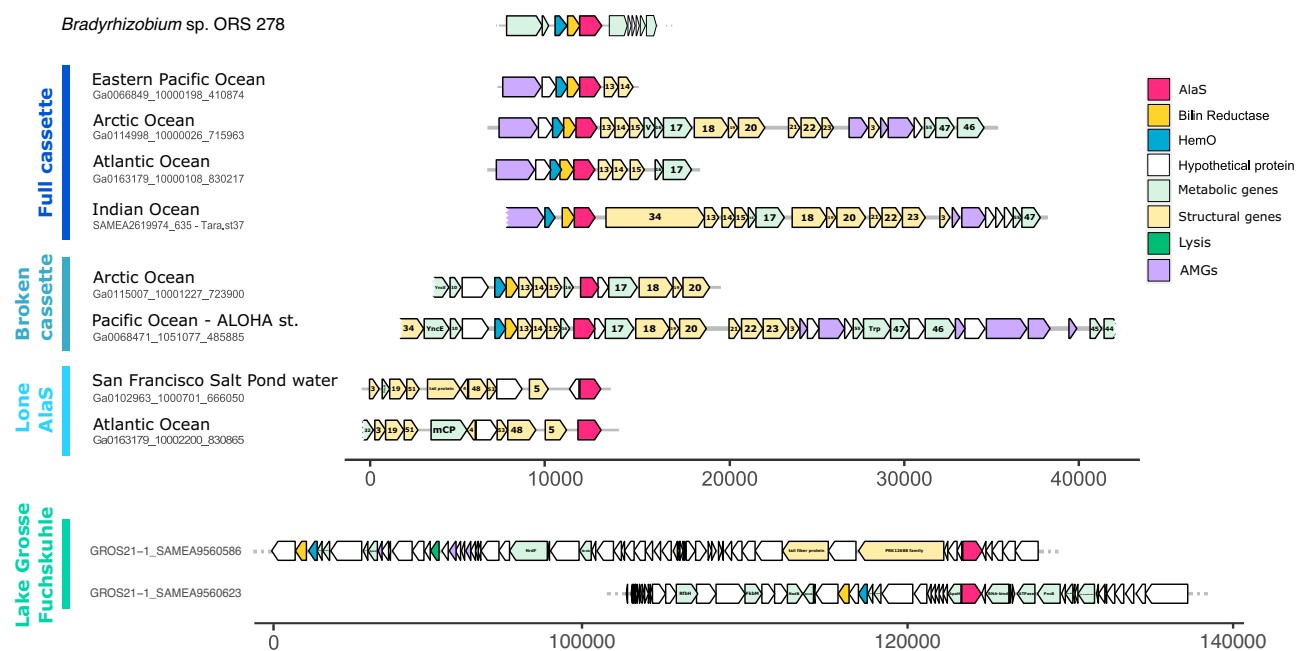

**Fig. 2 | Genomic contigs containing *valaS*.** Schematic representation of representative genomic contigs retrieved in this project. Genes are colored by categories according to the genome legend. Gene annotations are written within the gene when available; "gene products (gp)" are marked solely by their number. Contigs longer than presented are marked with a dotted line. The scale bar denotes bp. Sequences for the contigs can be found in Supplementary Data 4.

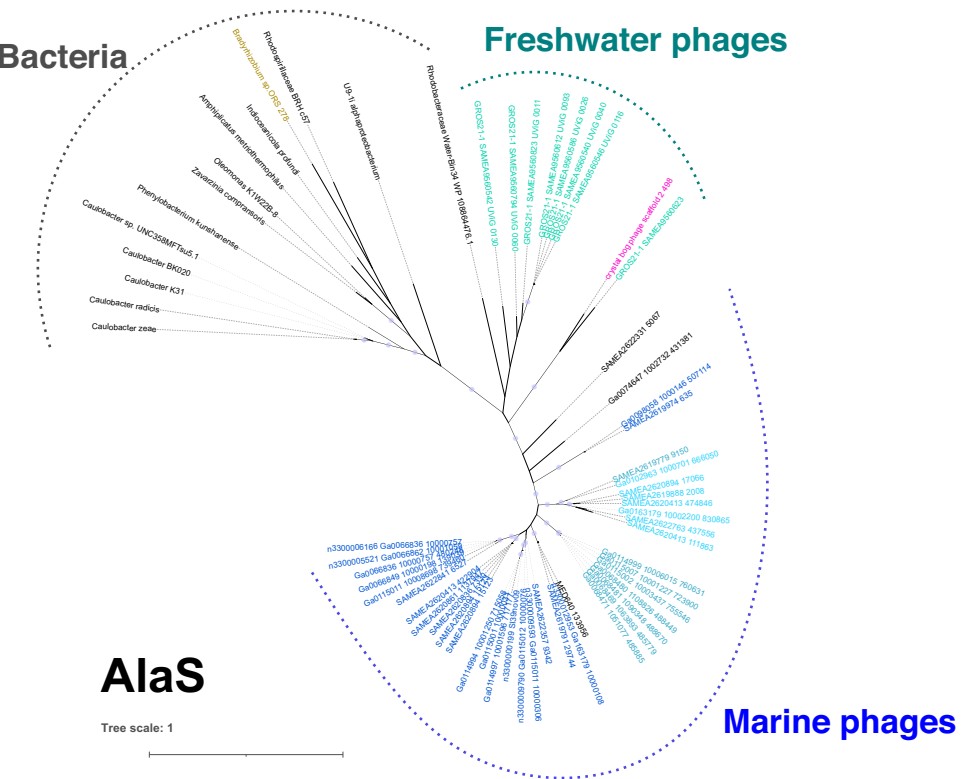

**Fig. 3 | Phylogenetic analyzes of AlaS protein sequences from bacterial and viral origin.** Unrooted maximum likelihood phylogenetic tree for AlaS. Black names denote sequences from bacteria and metagenomic contigs of uncertain origin. *Bradyrhizobium* sp. ORS278, containing the three gene-cassette, is marked in gold. The vAlaS sequence from CB_2 phage chosen for experimental characterization is colored pink. Metagenomically retrieved contigs from this project are color coded according to the bars in Fig. 2. Circles represent bootstrap values > 0.9. The scale bar indicates the average number of amino acid substitutions per site.

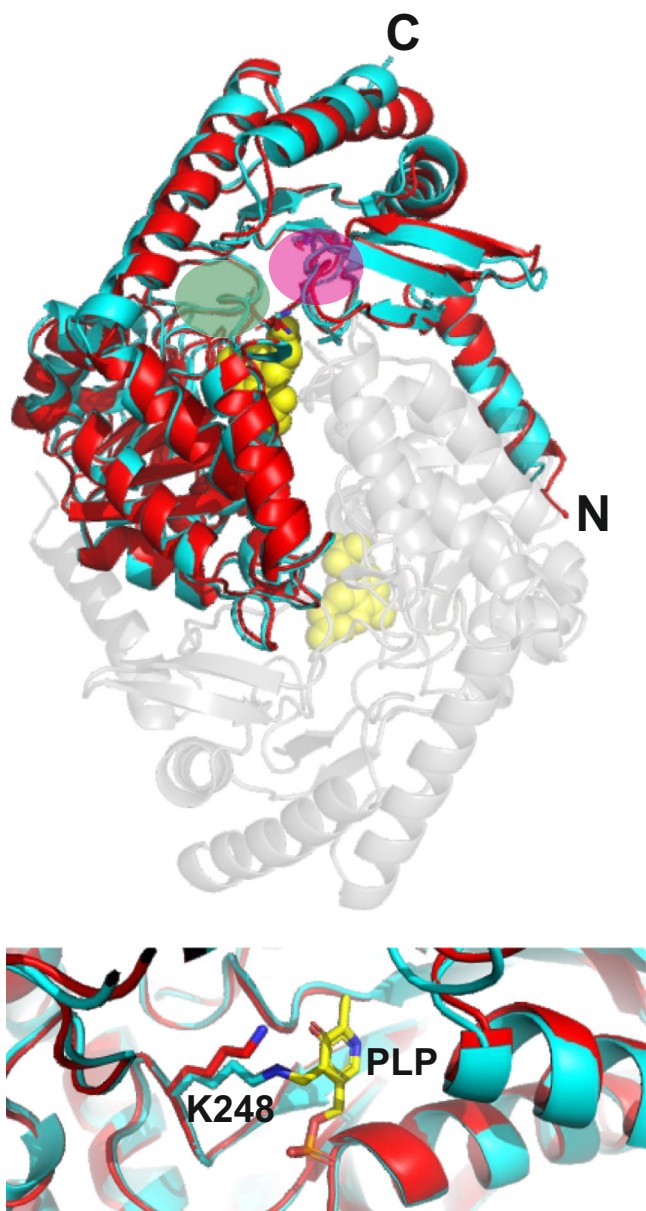

**Fig. 4 | Structural model of monomer vAlaS (CB_2) (red) and overlay with bacterial AlaS (blue) from *R. capsulatus* using AlphaFold.** The dimer structure of vAlaS is depicted transparently in conjunction with its second monomer. Yellow spheres represent the pyridoxal phosphate (PLP) cofactor, one on each monomer. Glycine binding site (green circle) and succinyl-CoA binding site (purple circle). The lower picture shows an enhanced view of the PLP binding site where a Schiff's base is formed with amino acid residue K248 (*R. capsulatus* numbering). A PDB file of the vAlaS modeled structure is available as Supplementary Data 7.

bacterial proteins (Supplementary Fig. S1). The FDBR sequences group according to the five FDBR families, with the metagenomically retrieved viral sequences clustering within PcyX. Interestingly, the PcyX sequence from the CB_2 phage groups midway between PcyX and PcyA, together with two bacterial FDBRs (Supplementary Fig. S2, PcyA*). Therefore, we asked whether the CB_2 encoded enzymes are functional. To confirm the biochemical activity, recombinant CB_2 vHemO and vPcyA/X were purified to almost homogeneity. Although, as expected, vHemO nicely converted heme to BV IXα, thus confirming it as an active heme oxygenase (Fig. 6a), the results for PcyA/X were rather surprising. To date, all characterized viral FDBRs that are in a two-gene cassette with *hemO* have been shown to belong to a new class

of FDBRs that we termed PcyX, as they have significant sequence homology to cyanobacterial (and other phages) PcyA. However, unlike PcyA, which converts BV in a four-electron reduction to PCB, PcyX synthesizes the PCB isomer PEB from its BV substrate (see also Fig. 1) and has thus far only been found to be present in viruses from marine environments[6,16]. Here, we show that PcyA/X of the freshwater phage CB_2 is a real PcyA that converts BV to PCB in a single reaction and can supply the chromophore to build a functional phytochrome photoreceptor (Fig. 6a, b). This activity can likely be explained by the PcyX specific catalytic residue Asp55 which is absent in the viral sequence. Overall, the occurrence of this three gene cassette from phage genomes is interesting with regard to a genomic island in the alphaproteobacterium *Bradyrhizobium* sp. ORS278, which contains the same three genes *hemO-pcyA* and *alaS*[26] (Fig. 2). To the best of our knowledge, this is the only bacterial example where the Shemin pathway *alaS* is found together with genes encoding phycobilins, raising the question of the origin of the viral copies found and characterized here.

## Discussion

### v*alaS* is a new AMG involved in tetrapyrrole biosynthesis

AMGs are widely distributed in the genomes of bacteriophages and are thought to improve the fitness of the phage in its so-called virocell stage[27]. This is the stage in which phages actively proliferate within the host cell and in which the demand for energy in the form of ATP and NADPH is high for the production of progeny. Furthermore, phages induce metabolic reprogramming of the virocell to meet the demand for precursors for protein and nucleic acid synthesis[10,28]. Host DNA is rapidly degraded and many host genes are negatively regulated, as shown for cyanophages and their hosts[29]. The viral-encoded vAlaS detected in our study could help maintain the production of heme and other tetrapyrroles, which are vital for retaining the respiratory electron transport chain and other tetrapyrrole-dependent metabolic processes such as vitamin $B_{12}$-dependent methylmalonyl-CoA mutase (MMCM)[30] in the virocell (Fig. 7). The latter is involved in the degradation of fatty acids and several amino acids, funneling the breakdown products into the tricarboxylic acid cycle[31]. In addition, vAlaS sequences were specifically found in peat bogs, an environment that is rich in bacteria that perform extracellular electron transfer utilizing multi heme *c*-type cytochromes[32,33]. The identification and functional characterization of a viral-encoded AlaS thus adds a new functional member to the wide variety of AMGs. Genomic reconstruction of both phage and host genomes from metagenomic data suggests that *alaS*-carrying phages exist either infecting alphaproteobacteria or non-alphaproteobacteria[13]. The CB_2 phage (AMGs analyzed within this work) is proposed to infect *Methylocystis* species[13], bacteria belonging to the alphaproteobacteria, and carry out methane oxidation to methanol for energy metabolism. Another v*alaS*-encoding phage, TP6_1, is proposed to infect *Methyloparacoccus* sp., a gammaproteobacterium[13]. Both organisms are non-photosynthetic and encode the Shemin and C5 pathways for ALA synthesis, respectively.

### vAlaS could be beneficial for phages infecting both alpha- and/or non-alphaproteobacteria

Phages carrying v*alaS* sequences would be able to maintain or even increase tetrapyrrole biosynthesis upon infection in various bacterial hosts independent of their own way of synthesizing 5-ALA. A phage carrying v*alaS* would have several advantages during the infection of a non-alphaproteobacterial host (i.e., gammaproteobacteria such as *Methyloparacoccus*): 1. Using the single Shemin pathway enzyme AlaS instead of two enzymes (GtrR and GsaM), as known for the C5 pathway, would reduce the genomic burden of the phage DNA packaging, similar to PebS[5] (where one new phage enzyme comprises the function of two host enzymes, catalyzing the same reaction). 2. Boosting/maintaining heme/tetrapyrrole biosynthesis to meet the demand for

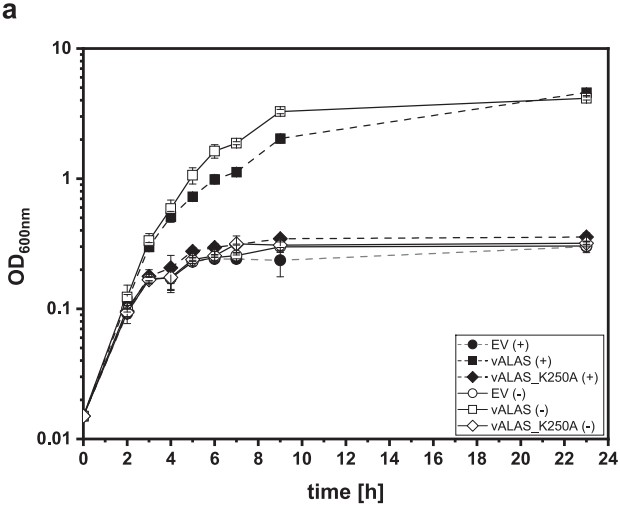

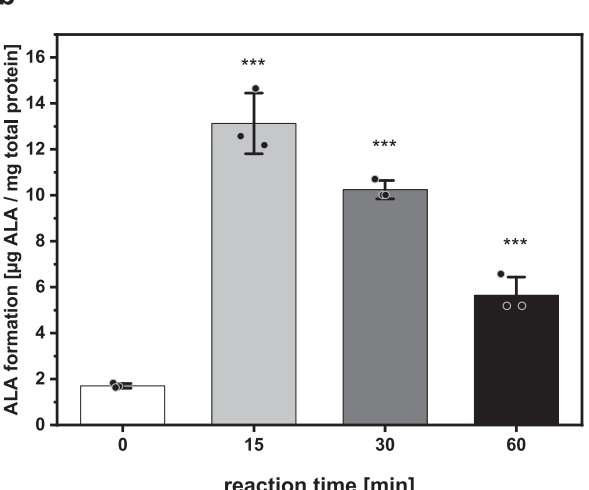

**Fig. 5 | Functional complementation of ALA-auxotrophic *E. coli* ST18 with a functional and non-functional vAlaS. a.** Growth of ST18 containing plasmid encoded vAlaS (squares), plasmid encoded vAlaSK250A variant (diamond) and the vector control (circles) in the presence (filled) or absence (empty symbol) of inductor IPTG. The data are mean values for three biological replicates for each strain and condition. Standard deviation is shown in error bars at each time point. **b.** vAlaS shows distinct enzyme activity in cell lysate when supplemented with C4 specific substrates. Cell lysate of complemented *E. coli* strains was supplemented with glycine and succinyl-CoA to detect product formation over time. Time points demonstrate the respective amount of ALA that was measured after subtracting the already present ALA in the cell (time point 0 min). The data shown represents the mean values of all measurements and the respective standard deviation. Measurements were conducted with three independent biological replicates and two technical replicates ($n = 6$). For statistical determination, values were tested for a normal distribution using the Shapiro–Wilk test, and normally distributed data were analyzed using one-way ANOVA with Bonferroni correction and paired t tests to determine the significance levels of each time point against time point 0. Significance levels were assigned accordingly: $p \leq 0.05$ equals *; $p \leq 0.01$ equals ** and $p \leq 0.001$ equals ***. Source data are provided as a Source Data file.

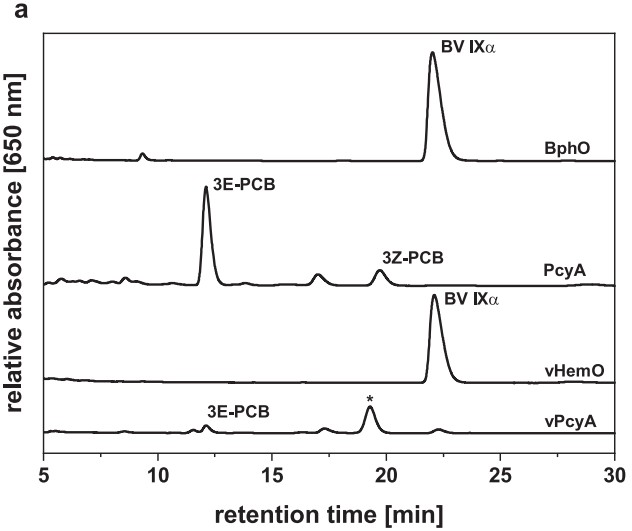

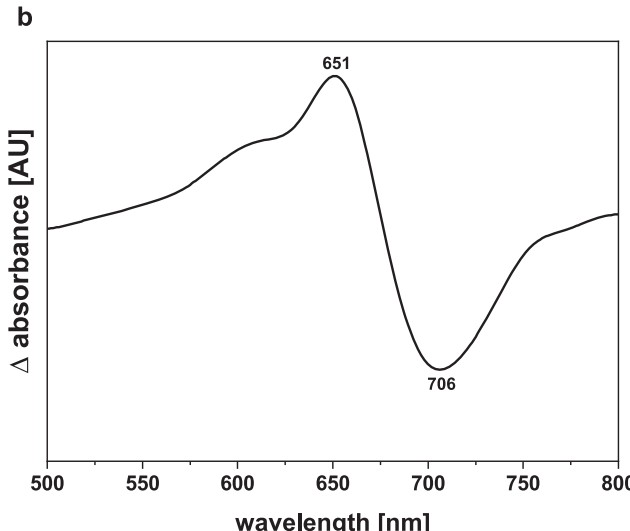

**Fig. 6 | Enzyme activity for the vHemO- and vPcyA-catalyzed reactions. a** HPLC analyzes of bilin reductase and heme oxygenase activity monitored at 650 nm. One reaction product of vPcyA is marked with an asterisk and could be the intermediate $18^1,18^2$-DHBV. BphO[70], heme oxygenase; PcyA, cyanobacterial PCB:ferredoxin oxidoreductase[67]. **b** Phytochrome difference spectrum was obtained following coexpression of genes encoding Cph1, vHemO and vPcyA and purification of Cph1 via metal affinity chromatography. In vivo chromophore assembly was detected after illumination with red light (636 nm−Pfr spectrum) and far-red light (730 nm for Cph1−Pr spectrum). Difference spectra were calculated by subtracting Pfr from the Pr spectrum. Source data are provided as a Source Data file.

tetrapyrrole cofactors in respiratory or other metabolic enzymes like MMCM. 3. Enabling host tRNA$^{Glu}$ to exclusively feed protein biosynthesis, which could be advantageous during infection (Fig. 7). Furthermore, based on bioinformatic analysis, the known heme binding site of AlaS for negative feedback inhibition is not present in vAlaS[34]. This would enable constant production of 5-ALA in the infected host.

The advantages for a *valaS*-carrying phage infecting an alpha-proteobacterial host are rather straightforward. The synthesis of 5-ALA, the precursor of all natural tetrapyrroles, is one of the crucial regulatory points in this metabolic pathway and is transcriptionally and posttranslationally tightly regulated by several environmental factors in bacteria, including oxygen, iron and heme[34,35]. It therefore makes sense to equip the phage with a *valaS* copy that is not prone

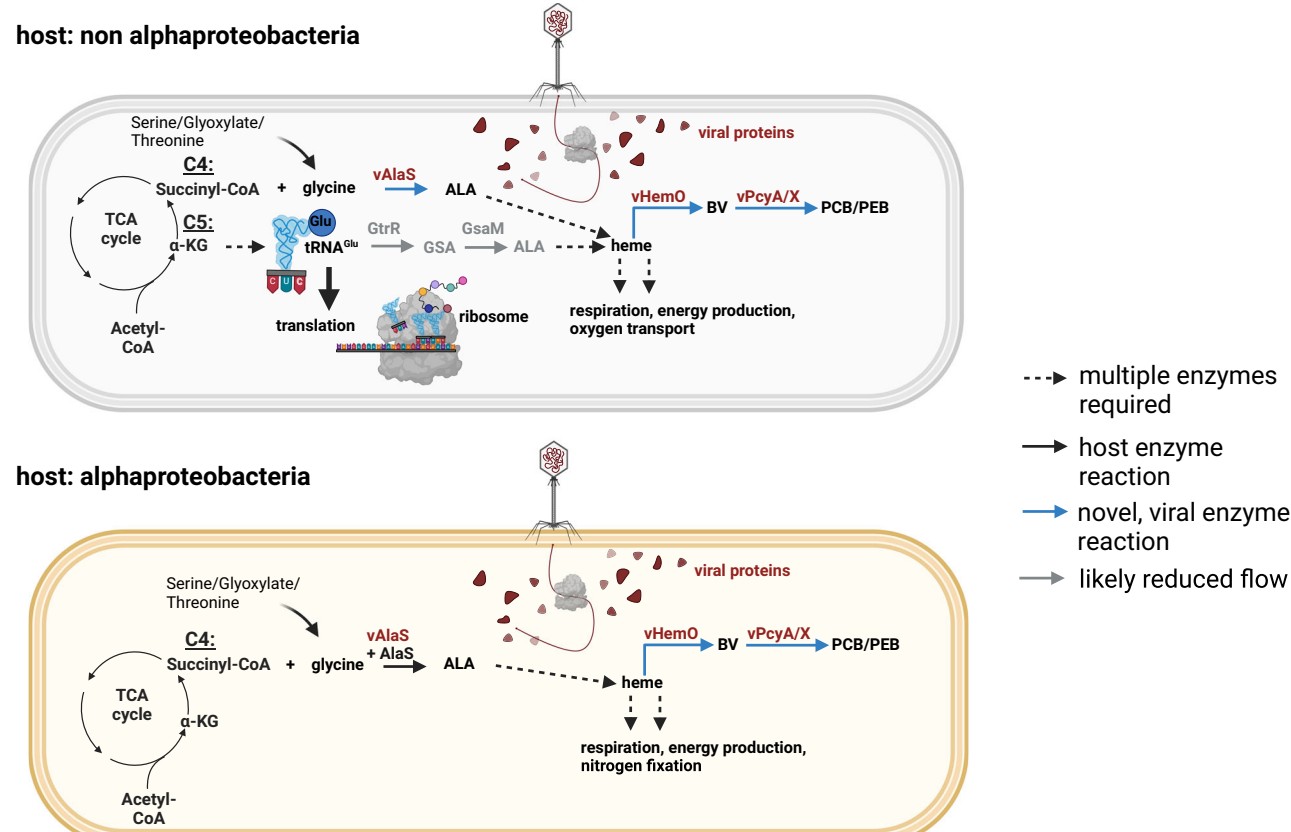

**Fig. 7 | Proposed model of phages carrying all three tetrapyrrole biosynthesis genes and their impact on alphaproteobacterial and non-alphaproteobacterial cells.** We hypothesize that in a non alphaprotobacterial host vALAS is responsible for tetrapyrrole biosynthesis during infection, enabling the virocell to employ glutamyl-tRNA for protein biosynthesis instead of tetrapyrrole biosynthesis. In alphaproteobacterial hosts, we propose that vAlaS supports its own AlaS in help of generating sufficient amounts of tetrapyrroles to support respiration and energy production. See text for more details. Created with BioRender.com, released under a Creative Commons Attribution-NonCommercial-NoDerivs 4.0 International license.

to feedback inhibition by heme or other types of regulation to ensure constant 5-ALA synthesis in the virocell to support the cell with tetrapyrrole precursor molecules. Independent of the host cell and its metabolism, tetrapyrroles are essential cofactors required for a number of enzymes that will help the virocell redirect metabolites for central metabolism and furthermore retain the functionality of heme-containing respiratory enzymes like cytochrome c oxidase.

### What is the purpose for tetrapyrrole catabolic enzymes in the virocell?

Still enigmatic is the co-occurrence of *hemO* and FDBR genes on some of the contigs containing v*alaS*, sometimes even as a three-gene cassette. Such a three-gene cassette has previously only been identified in the photosynthetic, plant-associated alphaproteobacterium *Bradyrhizobium* sp. ORS 278, which is, to our knowledge, the only alphaproteobacterium carrying this genomic island likely acquired by horizontal gene transfer[26]. This island includes *hemO*, *pcyA* and *alaS* genes with an adjacent phytochrome gene *bphP*. Together, the genes were shown to be capable of producing a functional phytochrome photoreceptor with a PCB chromophore[16,26].

The characterization of the marine AMG cassette consisting of *hemO* and *pcyX* related to pigment biosynthesis was previously carried out and is believed to ensure proper photosynthetic light reactions[6]. Genes encoding enzymes for heme oxygenase and bilin reductases are proposed to keep intact phycobilisomes to facilitate proper light-harvesting of the cyanobacterial virocell[5]. The findings within this work revealed that the cassette is not restricted to phages from the marine

environment but is also present in freshwater habitats. Furthermore, we show that the FDBR identified in the CB_2 phage genome is a true phycocyanobilin-ferredoxin oxidoreductase (PcyA), which converts BV to PCB, a chromophore to light-harvesting phycobiliproteins of cyanobacteria, or to the phytochrome photoreceptor. PCB was recently shown to be an important regulator of chlorophyll biosynthesis in oxygenic phototrophs by binding to GUN4 and thereby stimulating magnesium chelatase activity[36]. As the CB_2 phage is proposed to infect alphaproteobacteria, a function in light-harvesting is unlikely, as the phototrophic members of this genus use a bacteriochlorophyll-based antenna[37]. The phototrophic alphaproteobacteria also lack GUN4, but with BchJ possesses a homolog[36]. However, so far, it is not known whether PCB interacts with BchJ. Therefore, we can only speculate on a fitness advantage that *hemO* and *pcyA/X* genes could have for such a phage. In this regard, a role in iron supply during infection has also been discussed[6], and many of the metagenomic data are acquired from nutrient-poor environments[38], and iron is needed for electron transfer reactions in energy metabolism. The identified catalytic activity of freshwater-derived PcyA analyzed in this work that generates a functional phytochrome chromophore aligns with the recently proven ancestor of PcyA (prePcyA) that evolved in non-photosynthetic proteobacteria[39].

To date, this is the first heme oxygenase and PcyA from freshwater viruses whose function has been entirely demonstrated. Furthermore, the formation of a functional light-sensing phytochrome or the interaction with a bilin-binding globin protein, phycobiliprotein paralogs known as BBAGs (bilin biosynthesis-associated globins) from the produced PCB is one possibility[39]. Future advanced phage-host

isolation will hopefully enable us to understand the role of these new interesting AMGs and their importance for phage infection.

## Methods

### Metagenomic data analysis

Metagenomic datasets from the *Tara* Oceans were reassembled as described elsewhere[17,40]. Viral fatty acid desaturase sequences obtained in a previous study[17] were used as queries using TBLASTX[41,42] with default parameters to recruit scaffolds containing similar genes in the reassembled *Tara* Ocean database and in publicly available datasets from the JGI GOLD[43]. One scaffold, likely originating from a pelagiphage (based on gp13 phylogeny), encoded a fatty acid desaturase, an FDBR, HemO and the vAlaS. We used the viral encoded AlaS as bait in the same databases following the same procedure to retrieve more relevant contigs.

ORFs in the contigs were identified using GeneMark[44,45] and manually annotated using BLASTX with default parameters against the NCBI nonredundant (nr) protein database to identify and annotate the putative genes in the contigs. These annotations were validated using the NCBI conserved domains[46] and curated manually. Genomic maps were constructed using Geneious Prime® v2023.2.1, ggplot[47] and Inkscape v1.2.2.

For the time series data from German lakes (LIMNOS), four lakes in the vicinity of Berlin were sampled between 2003 and 2020, resulting in 1096 metagenome samples in total. Samples have been sequenced and assembled individually (Supplementary Data 2). Contigs of length at least 500 bp were first searched with VIBRANT[48] for virus signals. All the hits were then subjected to quality assessment with CheckV[49], and only virus sequences of medium or high quality were retained. The resulting 50,913 viral contigs were annotated with geNomad[50] and searched with Diamond v2[51] against a protein database of previously detected AlaS sequences. All twelve contigs with a hit were retained for further analysis.

### Relative abundance assessment

To get a better assessment of vAlaS abundance in the environment we screened the *Tara* Oceans viromes[52] and our German Lakes (LIMNOS) datasets against the vAlaS and vPsbA protein sequences using DIAMOND[51] with the following paramenters: −very-sensitive −strand both −query-cover 60 −outfmt 6 qseqid sseqid pident length mismatch gapopen qstart qend sstart send evalue qcovhsp scovhsp full_qqual qstrand -k. We then filtered out all results with less of 70% identity. The counts can be found in Supplementary Data 3.

### Phylogenetic construction and analyses

Genes encoding FDBRs, HemO, AlaS and the neck-protein gp13 were translated into proteins and aligned along sequences from isolated phages and their bacterial hosts (Supplementary Data 6). Maximum likelihood phylogenetic trees were created using the Phylogeny.fr pipeline[53,54], including the MUSCLE alignment[55], PhyML v3.0[56], and the WAG substitution model for amino acids[57]. One hundred bootstrap replicates were performed for each tree, and only bootstraps above 0.9 are shown as circles in the trees. See Supplementary Data 6 for the sequences used to create the trees. gp13 was chosen for tracing the phylogeny of the phages since it is a single copy gene found solely in phages with a *Myoviridae* phenotype, it is a structural component of the capsid, is well conserved, and abundant in our long and short metagenomic contigs[58].

### Structural modeling

Prediction of the 3-D structure of vAlaS from CB_2[13] was generated using AlphaFold2[59], with ColabFold[60], by utilizing the AlphaFold2 advanced notebook on GitHub. For modeling the multiple sequence alignment (MSA) method mmseqs2 (max_msa 512:1024) was chosen for a 1:1 homooligomer, the number of models were five and the maximal recycles were three by one sample. The unpaired mode was chosen and without trimming of the sequence. The chosen model regarding pLDDT (96.11) has am pTMscore of 0.9439. Overlays were created utilizing PyMOL v.2.0[61] of vAlaS and bacterial proteins (*Rhodobacter capsulatus*; RcA) with bound substrates and/or cofactors (PDB code: 2BWN, 2BWO, 2BWP)[18]. The PDB file of predicted vAlaS structure is available as Supplementary Data 7.

### Functional complementation of 5-ALA-auxotrophic *E. coli* Δ*gtrR* with plasmid encoded vAlaS

To assess the in vivo activity of vAlaS, an *E. coli* codon-adapted version of the *valaS* gene from the Crystal_bog_pmoC-phage_2 (CB_2)[13] was inserted into the pCYB1 vector (New England Biolabs) under the control of a *tac* promotor and omitting a gene fusion to the vacuolar ATPase subunit (VMA) intein sequence. Codon adaptation was optimized using the JGI BOOST codon adaptation tool[62] with settings 'balanced' and then chemically synthesized (Twist Bioscience). The non-functional variant vAlaS K250A was generated by QuikChange mutagenesis on the synthetic gene sequence. Both constructs were used for complementation studies in the 5-ALA-auxotrophic *E. coli* ST18 strain[22]. This strain contains a deletion of the *gtrR* gene (formerly *hemA*) and is auxotrophic when not supplemented with 5-ALA [50 μg/mL] (Sigma-Aldrich; A3785). As positive controls, the vector pRcA consisting of pGEX-6P with the *alaS* gene of *R. capsulatus*[19] and the vector pMkA of pGEX-6P with the *gtrR* gene of *Methanopyrus kandleri*[63] encoding a glutamyl-tRNA reductase (GtrR) were selected and transformed into auxotrophic *E. coli* ST18[22]. The respective empty vectors were used as controls. Freshly prepared chemically competent *E. coli* ST18 cells were used for functional complementation. During the preparation of chemically competent cells, after heat shock recovery and initial plating, the addition of 50 μg/mL 5-ALA was ensured.

### Functional complementation growth experiment

Cultivation was conducted in a sterile 100 mL glass flask filled with 20 mL LB medium and the respective antibiotics. The inoculum was an overnight culture from the transformation plates supplemented with ampicillin and 5-ALA. Before inoculation, the cells were washed twice in LB medium without 5-ALA and then used to adjust the start $OD_{600 nm}$ (either 0.015 or 0.03). During incubation at 37 °C with shaking at 180 rpm, the flasks were fully covered in aluminium foil to prevent light exposure. During growth, gene expression was induced by the addition of 0.5 mM IPTG (Sigma-Aldrich; I5502) after the cultures reached an $OD_{600 nm}$ of 0.1–0.15. After incubation for 24 h, the cultures were serially diluted and plated on agar plates containing antibiotics and the presence/absence of IPTG to determine functional complementation following incubation at 37 °C overnight. The formation of colonies indicated successful complementation, and empty vector controls were also plated and showed no colony formation. All experiments were conducted as independent biological triplicates with technical duplicates. The growth curve data represent the mean values of all measurements and the respective standard deviation.

### Internal 5-ALA measurement

Fifty millilitres of complemented *E. coli* ST18 was grown as stated before but in a sterile 250 mL glass flask for 24 h. All cultures were grown in the presence of IPTG as stated before. For harvesting the cells, the culture was centrifuged for 20 min at 3,700 x g at 4 °C. The pellet was resuspended and washed two times with 50 mM Tris-HCl buffer pH 7.5 to eliminate external 5-ALA contamination. For internal 5-ALA formation measurements, the pellets were resuspended in 500 μl lysis buffer (50 mM Tris-HCl buffer, pH 7.5) for each 100 mg of wet cell weight. The cells were disrupted using sonication (Ultrasonic homogenizer, Bandelin Sonoplus HD 2200 with tip KE 76) in an ice-water bath for a total of 1.5 min sonification with intervals of 30 s sonification and 1 min cool down. The sonication pulse was 40 %

intensity, 5 x cycles. Cell debris was spun down for 30 min at $17,000 \times g$ and 4 °C. The supernatant was removed, and protein concentration was determined using the Bradford reagent (Roti®Quant, Carl Roth), as recommended by the manufacturer, in transparent 96-well plates (Sarstedt, PS, flat based). Protein concentrations were determined using a calibration curve with BSA solubilized in the same buffer and measured in a plate reader (Tecan Infinite® F200 Pro) at 595 nm[64]. For the assays, the protein content was set to 2 mg/mL, and substrates of AlaS (glycine (Sigma-Aldrich; G7126), succinyl-CoA (Sigma-Aldrich; S1129) and cofactor (PLP; Sigma-Aldrich; P9255) were added to measure product formation (5-ALA) over time. In a 2-mL reaction tube, a 1-mL approach in 50 mM Tris-HCl buffer (pH 7.5) was carried out with 5 mM glycine, 0.1 mM succinyl-CoA and 0.5 mM PLP. The reaction was started by the addition of the cell lysate (2 mg/mL) and incubated at 37 °C on a heating block for different time points (0 min, 15 min, 30 min, 60 min). The reaction was terminated by adding 0.5 mL of 10 % TCA (w/v). Following a centrifugation step at 4 °C and $17,800 \times g$ for 5 min, the supernatant was mixed in a test tube with 2 mL of 1 M sodium acetate buffer (pH 4.7). Pyrrole formation was achieved by mixing the reaction mixture with 50 μL acetylacetone (Sigma-Aldrich, P7754), followed by boiling at 100 °C for 15 min. After cooling, 3.5 mL of modified Ehrlich reagent[65] were added, and pyrrole formation was measured at 554 nm after 5 min of incubation. The extinction coefficient of the ALA-pyrrole $66,000 \text{ M}^{-1} \text{ *cm}^{-1}$ [66] was used to determine 5-ALA formation with regard to protein content (mg of total protein). Time point zero represents the initial 5-ALA concentration in the cell lysate and was used as a blank to calculate only new 5-ALA formation. The data shown represents the mean values of all measurements and the respective standard deviation. Measurements were conducted with three independent biological replicates and two technical replicates. For statistical determination, values were tested for a normal distribution using the Shapiro–Wilk test, and normally distributed data were analyzed using one-way ANOVA with Bonferroni correction and paired t tests to determine the significance levels of each timepoint against timepoint 0. Significance levels were assigned accordingly: $p \leq 0.05$ equals *; $p \leq 0.01$ equals ** and $p \leq 0.001$ equals ***.

## UV–Vis and fluorescence measurements
The measurements were conducted in quartz cuvettes using an Agilent 8453 series spectrophotometer and Jasco FP-8300 fluorometer with the protein concentration set to 2 mg/mL for porphyrin accumulation measurement. Absorption curves were baseline subtracted using Origin 2022 software. Fluorescence readings were performed by using an excitation at 400 nm and emission at 415 nm to 750 nm. Excitation and emission bandwidths were each 5 nm and response time was 50 msec at medium sensitivity.

## Heterologous expression, purification and characterization of vHemO and vPcyA/X
Codon-adapted genes (see above) of vhemO and vpcyA/X from the CB_2 phage were cloned in pGEX-6P-3 (GE Healthcare Life Sciences), resulting in a tac promoter-driven expression of a glutathione-S-transferase fusion protein with the protein of interest. Expression and purification were performed as described previously[67] using E. coli BL21 (DE3) (vhemO) or E. coli Rosetta (DE3) pLysS (vpcyA/X) as a bacterial host. Heme oxygenase and anaerobic bilin reductase activity assays were performed as described previously with minor modifications[6,68]. Bilin products were analyzed using an Agilent 1100 series chromatograph equipped with a Luna® 5 μm reversed-phase C18-column (Phenomenex, Torrance, California, USA) and a photodiode-array detector as described before. The mobile phase consisted of 50 % (v/v) acetone (Carl Roth, 7328.2) and 50 % (v/v) 20 mM formic acid (Carl Roth; 1EHK.1) at a flow rate of 0.6 mL/min. Reaction products were identified by comparison of the retention time

with known standards as well as whole spectrum analysis of the elution peaks. Coexpression of vpcyA (MCS1) together with vhemO (MCS2) from phage CB_2 was achieved by cloning each in one of the multiple cloning sites of pACYCDuet (Novagen) and coexpressing it with a T7 promoter-controlled cyanobacterial phytochrome containing a C-terminal His$_6$-tag in E. coli[69]. In PBS buffer (pH 7.4), partially purified cyanobacterial phytochrome Cph1 via TALON® (Cytiva) affinity chromatography was analyzed regarding its absorbance spectra, which were recorded after incubation for 3 min with red light (636 nm–Pfr spectrum) and after incubation for 3 min with far red light (730 nm–Pr spectrum). Pfr/Pr-difference spectra were calculated by subtracting Pfr from the Pr spectrum. The curve was smoothed using an FFT filter in Origin 2022 software.

## Reporting summary
Further information on research design is available in the Nature Portfolio Reporting Summary linked to this article.

## Data availability
All genomic contigs analyzed in this paper can be found in the Supplementary Data 1. Links to publicly available JGI/NCBI datasets, as well as the unpublished LIMNOS datasets analyzed in this study can be found as Supplementary Data 2. The LIMNOS metagenomic datasets (BioProject PRJEB47226) will become publicly available upon publication of a separate manuscript (expected ~ 12/2024). In the meantime, access to the LIMNOS datasets can be obtained from HPG (hanspeter.grossart@igb-berlin.de), without restrictions. Source data are provided with this paper.

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

## Acknowledgements

This work was supported by a grant from the Deutsche Forschungsgemeinschaft within the priority program 2330 (to N.F.D.), by European Commission ERC Advanced Grant 321647 (O.B.), and Israel Science Foundation grant 143/18 (O.B.). S.R. was funded in part by the Ariane de Rothschild Women's Doctoral Program and an EMBO post-doctoral fellowship. We would like to thank Shinichi Sunagawa and Hans-Joachim Ruscheweyh for sharing unpublished data and for bioinformatic support. J.W. and H.P.G. thank Leibniz-IGB and all technicians at Dept. Plankton and Microbial Ecology at IGB for financial and sampling support, respectively. Additional thanks go to the Banfield lab for sharing sequences and Dieter Jahn and Jürgen Moser for sharing plasmids and the *E. coli* ST18 strain. Technical support by Elena Monzel and Eugenio Perez Patallo is acknowledged. We thank Steffen Heck for helpful discussions, and Haim Ashkenazy for helping with bioinformatic analyzes.

## Author contributions

H.W., S.R., O.B., S.Z., and N.F.D. designed the research, S.R. and A.K. performed bioinformatic analysis, J.W. and H.P.G. contributed unpublished metagenomic datasets, H.W., S.Z., and V.B. performed the experiments, H.W. and N.F.D. wrote draft of the manuscript with the help of all other authors.

## Funding

## Competing interests

The authors declare no competing interests.
