## [Peer Review File · Nature Communications]

Identification of Shemin pathway genes for tetrapyrrole biosynthesis in bacteriophage sequences from aquatic environmentsReviewer #1 (Remarks to the Author):

The work entitled "Virus-encoded Shemin pathway highlights the importance of tetrapyrrole metabolism during host infection" by Wegner et al. discovered 5-ALA synthase genes of the Shemin pathway in bacteriophage genomes retrieved from both marine and freshwater environments. Viral *alaS*s (*valaS*) are predicted to be functional and are proved to be active in a 5-ALA-auxotrophic *Escherichia coli* strain. Together with the previous identification of viral heme-derived pigment biosynthesis genes (Ledermann et al., 2016), the discovery of *valaS*s in this study provides new insights into the viral way of reprogramming host energy metabolism. This work is interesting and is important for furthering our understanding of the wide variety of AMGs. I have several concerns on this manuscript.

- 1) It is necessary to calculate the relative abundance of the *valaS* in viral communities by blast against reads data of the viromes. To assess its importance in aquatic ecosystems, the relative abundance of *valaS* can be compared to that of *psbA* or other important AMGs.
- 2) More analyses can be made to predict the hosts of phages bearing *valaS*.
- 3) Lines 303-304, It is not accurate to use "abundant" to describe the presence of *valaS* in the LIMNOS German lake without data support.
- 4) Lines 476-477, Since only four Berlin lakes were searched in this study, "widely distributed in freshwater" is overexpressed.
- 5) More description on *gp13* is needed, such as "mainly of myovirus origin".
- 6) Lines 137-139, searching against the NR database is not enough for the annotation work, since many descriptions of protein sequences in NR are not accurate. Conserved domain identification is more helpful and accurate and can be used to assist gene annotation.
- 7) It is necessary to include a table in the supplementary material to interpret the information of contigs bearing *valaS*, such as contig length, source, presence of *vhemO* and *vpcyA/X*.

Reviewer #2 (Remarks to the Author):

Wegner et al. characterise a new viral auxiliary metabolic gene, *valaS*, a homologue of 5 aminolevulinic acid synthase. They show the viral enzyme can complement an *E. coli* mutant deficient in 5-ALA synthesis, and show 5-ALA synthase activity in cell extracts. Overall, this is an exceptionally well conducted study, with thoroughly described methods and well-presented results. I recommend this manuscript for publication with only few minor corrections to the text below.

Line 299: Not clear how these sequences resemble those in Mosig pelagiphage. Could the authors quantify this?

Lines 310-314: Could this be clarified. As written it implies the Pelagibacteraceae are the only group using the Shemin pathway, rather than alphaproteobacteria. Maybe change 'only bacterial group' to 'only bacterial class', or reorder to make alphaproteobacteria the subject of the sentence.

Line 324: Replace 'nicely' with some quantification of the predicted structural fit.

Line 423: Not clear who is proposing the CB_2 infects *Methylocystis*. Or, more importantly how. Could the authors clarify this?

The discussion is well written and comprehensive, but is rather focussed on the metabolism aspect of *valaS* function in infection. Is there any evidence that 5-ALA synthesis might be involved in phage defence (e.g. through signalling). Is this worth discussing?

Figure 3: This could be better annotated to more clearly show what these groups are, without the reader needing to read the legend. Additionally could the authors detail how (if) this tree has been rooted. Lastly, while a partial amino acid alignment is presented in the supplementary, it would be useful if the authors could provide a full amino acid alignment used to construct the tree.

Reviewer #3 (Remarks to the Author):

This work builds on previous successes from these workers in elucidating a role for “rewiring” tetrapyrrole metabolism during phage infection of bacterial cells. Past work from Prof. Frankenberg-Dinkel and colleagues first identified enzymes for conversion of heme into bilins (linear tetrapyrroles). This included identification of functional sequences from cyanophages for heme oxygenase (HO, converting heme into biliverdin IX-alpha or BV), PcyA (converting BV into phycocyanobilin or PCB), and PebS (converting BV into phycoerythrobilin or PEB). PcyA and PebS are both ferredoxin-dependent bilin reductases (FDBRs), an enzyme family most widely known for its roles in synthesizing bilins for plant phytochrome signaling and for cyanobacterial light harvesting. More recently, the Frankenberg-Dinkel and Beja labs identified an additional FDBR, PcyX, in phages that do not infect cyanobacteria. PcyX carries out the same reaction as PebS, but the host cell is apparently not able to use the resulting PEB for light harvesting and energy metabolism. The current work extends these studies in several notable ways. First, the authors show that HO and PcyX are usually associated with a virally encoded enzyme for synthesis of delta-aminolevulinic acid (ALA) via the Shemin or C4 pathway. This viral ALA synthase (vALAS) is shown to be functional both in extracts and via rescue of an *E. coli* strain lacking the equivalent enzymatic function. The associated HO and FDBR sequences are also shown to be functional, but the reaction product of the FDBR is PCB rather than PEB.

Overall, this work thus provides valuable insights. The presence of virally encoded enzymes for the committed step in tetrapyrrole biosynthesis extends the previous work from these groups and implicates an important role for tetrapyrrole biosynthesis in phage infection, even in the absence of photosynthesis by the host cell. Moreover, the demonstration of PcyA activity for this particular viral FDBR seems surprising based on the phylogenetic analysis presented in this work. This result thus provides new insight for the FDBR field and also may facilitate improved biosynthesis of bilins in heterologous systems, a growing field in its own right. The overall experimental quality is good, and the experimental methodology is very well described. To this reviewer, there are only a few points that could be considered in a revised version.

1. The apparent lack of any promoter induction could perhaps be described as “constitutive” rather than “leaky.” The demonstration of ALAS activity is thus quite important in interpreting the complementation result in *E. coli*. Have the authors considered drawing on the extensive library of known loss-of-function ALAS mutations? Expressing an inactive form of vALAS would be a very valuable addition to this line of experiments, because it should not rescue this strain no matter how leaky the promoter is.

2. Can the light sensitivity of the porphyrin-producing strain be rescued by the co-expression of the viral HO, with or without the FDBR? This experiment could also provide insight into the

potential value of such proteins during phage infection: such proteins could provide a means of rewiring the cell to produce more heme while avoiding potential phototoxicity.

3. The authors provide convincing evidence for the presence of the first step in the Shemin (C4) pathway in some phages, encoded by vALAS and coupled with enzymes for production of bilins from heme (HO and PcyA/PcyX). They argue that increased flux through the heme biosynthesis pathway facilitates infection and that the Shemin pathway would be favorable relative to the Beale (C5) pathway because it requires a single enzyme to generate ALA and does not deplete charged Glu-tRNA molecules that would be useful for protein synthesis during infection. Have the authors looked for the presence of GluTR in phage genomes, with or without HO and FDBRs? This seems feasible to test, were GluTR to be in a cassette with FDBRs.

4. The characterized FDBR seems quite close to known PcyX enzymes, despite its clear PcyA activity. Can the authors provide any insight into the mechanistic basis for this change in regiospecificity? Are there unusual active-site residues or some other explanation?

5. The authors are effectively proposing that PCB can be supplied by phage-encoded PcyA to endogenous phytochrome photoreceptors, here in *Pelagibacter* spp. However, most bacterial phytochromes use BV as chromophore. Do the genomes of the proposed host cells encode candidate phytochromes that could incorporate a PCB precursor, as in cyanobacteria or *Bradyrhizobium* sp. ORS278?

Minor points:

1. Adding the *Bradyrhizobium* cassette to Fig. 2 would be useful for the reader, given that those sequences are highlighted as references in the trees. Also, what form of IMG accession is on this figure? It does not look like a scaffold code.

2. Adding peak wavelengths to Fig. 6B would be helpful.

Point-by point response to reviewer comments (author responses in italics):

Reviewer #1 (Remarks to the Author):

The work entitled "Virus-encoded Shemin pathway highlights the importance of tetrapyrrole metabolism during host infection" by Wegner et al. discovered 5-ALA synthase genes of the Shemin pathway in bacteriophage genomes retrieved from both marine and freshwater environments. Viral *alaS*s (*valaS*) are predicted to be functional and are proved to be active in a 5-ALA-auxotrophic *Escherichia coli* strain. Together with the previous identification of viral heme-derived pigment biosynthesis genes (Lederhann et al., 2016), the discovery of *valaS*s in this study provides new insights into the viral way of reprogramming host energy metabolism. This work is interesting and is important for furthering our understanding of the wide variety of AMGs. I have several concerns on this manuscript.

1) It is necessary to calculate the relative abundance of the *valaS* in viral communities by blast against reads data of the viromes. To access its importance in aquatic ecosystems, the relative abundance of *valaS* can be compared to that of *psbA* or other important AMGs.

Following this suggestion, we compared the relative abundance of vAlaS and vPsbA in marine and freshwater databases. We used the viromes of the Tara Ocean (Brum et al., 2017) and our Berlin Lake databases. Instead of using blast we calculated the abundance using protein sequences to recruit translated reads, to increase the sensitivity with DIAMOND. We found that in marine samples PsbA is 60 - ~175,000 times more abundant than vAlaS, suggesting that these are rare phages. In the freshwater samples vPsbA is ~5 to ~550 more abundant than vALAs. We added these results to the text in lines 137-140 .

2) More analyses can be made to predict the hosts of phages bearing *valaS*.

Indeed, determining the host of these uncultured phages is highly interesting. Unfortunately, the current computational methods for host prediction are limited and we can currently not obtain reliable hosts for these phages. For example, from the 77 phages shown in Fig. 3, only 19 have a prediction at the host genus level with iPHoP (<https://journals.plos.org/plosbiology/article?id=10.1371/journal.pbio.3002083>) and these results cover 5 different phyla (Verrucomicrobiota, Bacteroidota, Bacillota, Pseudomonadota, Patescibacteria). We thus do not have sufficient confidence in these results to mention them in the manuscript.

3) Lines 303-304, It is not accurate to use "abundant" to describe the presence of *valaS* in the LIMNOS German lake without data support.

We replaced the word "abundant" for "were also found".

4) Lines 476-477, Since only four Berlin lakes were searched in this study, "widely distributed in freshwater" is overexpressed.

We modified the sentence to better state that the cassette is found in both marine and freshwater environments, without referring to the abundance or global spread.

5) More description on *gp13* is needed, such as "mainly of myovirus origin".
*We described this more accurately in the methods and results, emphasizing that *gp13* is found only in myophages. Lines 125-128. Additionally, we further explained our choice of *gp13* as a marker gene in the methods (lines 361-363).*

6) Lines 137-139, searching against the NR database is not enough for the annotation work, since many descriptions of protein sequences in NR are not accurate. Conserved domain identification is more helpful and accurate and can be used to assist gene annotation.

*While doing the annotation work we validated our findings (obtained with BLAST against nr) with domain identifications tools (using the ncbi domain finder). We curated our contigs manually to prevent miss-annotations or omissions. This was, in fact, the way we actually found the first *valaS* gene, which was found by domain-identification and not by homology. We made it clearer in the text. Lines 365-369.*

7) It is necessary to include a table in the supplementary material to interpret the information of contigs bearing *valaS*, such as contig length, source, presence of *vhemO* and *vpcyA/X*.
A table including the contig's name, length, %GC, sample source and presence of the relevant genes in a cassette (Y/N) is now included, as Supp. Table S1

Reviewer #2 (Remarks to the Author):

Wegner et al. characterise a new viral auxiliary metabolic gene, *valaS*, a homologue of 5 aminolevulinic acid synthase. They show the viral enzyme can complement an *E. coli* mutant deficient in 5-ALA synthesis, and show 5-ALA synthase activity in cell extracts. Overall, this is an exceptionally well conducted study, with thoroughly described methods and well-presented results. I recommend this manuscript for publication with only few minor corrections to the text below.

Line 299: Not clear how these sequences resemble those in Mosig pelagiphage. Could the authors quantify this?

We added a reference to the relevant figures in the supplement, phylogenetic trees of HemO and FDBRs, where the proteins from Mosig phage are clustered within the proteins from our new contigs. Mosig is the only cultured phage clustering with our contigs in these proteins. When comparing structural proteins, like gp13, the two (only) cultured phages infecting Pelagibacter, HTVC008M and Mosig are the only phages clustering with our contigs, suggesting a high resemblance.

Lines 310-314: Could this be clarified. As written it implies the Pelagibacteraceae are the only group using the Shemin pathway, rather than alphaproteobacteria. Maybe change 'only bacterial group' to 'only bacterial class', or reorder to make alphaproteobacteria the subject of the sentence.

The sentence was reworded to make clear that all alphaproteobacteria utilize the Shemin pathway. The sentence (after formatting of document) can now be found on line 145-146.

Line 324: Replace 'nicely' with some quantification of the predicted structural fit.

We changed the wording and added the calculated RMSD value of 0.875 Å obtained from the overlay in Pymol.

Line 423: Not clear who is proposing the CB_2 infects Methylocystis. Or, more importantly how. Could the authors clarify this?

It was proposed by Chen et al, 2020 that CB_2 infects Methylocystis. The reference was added.

The discussion is well written and comprehensive, but is rather focussed on the metabolism aspect of *valaS* function in infection. Is there any evidence that 5-ALA synthesis might be involved in phage defence (e.g. through signalling). Is this worth discussing?

*Thank you for the comment and this interesting suggestion. We have no indication that *valaS* would be involved in phage defense. We suspect solely a metabolic function and therefore decided not to discuss a possible role in phage defense.*

Figure 3: This could be better annotated to more clearly show what these groups are, without the reader needing to read the legend. Additionally, could the authors detail how (if) this tree has been rooted. Lastly, while a partial amino acid alignment is presented in the supplementary, it would be useful if the authors could provide a full amino acid alignment used to construct the tree.

Labels were added to the tree to better show the origin of the proteins.

*This tree is un-rooted. We wanted to show the clustering of these proteins, since it clearly shows their organism of origin (viral/bacterial, freshwater/marine), without making any further assumptions. Based on this phylogeny, we cannot assess where was the viral version of *AlaS* acquired from, and whether it was later transferred horizontally to bacteria. We don't think it correct to make such assumptions on the origins and evolution of the protein without further knowledge on, at least, the hosts of these phages. Therefore, an un-rooted tree will show the different types of closely related proteins to the phage-encoded *AlaS*, without misleading, by forcing an outgroup. We modified the figure legend to state this is an unrooted tree.*

We added a full alignment file Supplementary File S2.

Reviewer #3 (Remarks to the Author):

This work builds on previous successes from these workers in elucidating a role for "rewiring" tetrapyrrole metabolism during phage infection of bacterial cells. Past work from Prof. Frankenberg-Dinkel and colleagues first identified enzymes for conversion of heme into bilins (linear tetrapyrroles). This included identification of functional sequences from cyanophages for heme oxygenase (HO, converting heme into biliverdin IX- α or BV), PcyA (converting BV into phycocyanobilin or PCB), and PebS (converting BV into phycoerythrobilin or PEB). PcyA and PebS are both ferredoxin-dependent bilin reductases (FDBRs), an enzyme family most widely known for its roles in synthesizing bilins for plant phytochrome signaling and for cyanobacterial light harvesting. More recently, the Frankenberg-Dinkel and Beja labs

identified an additional FDBR, PcyX, in phages that do not infect cyanobacteria. PcyX carries out the same reaction as PebS, but the host cell is apparently not able to use the resulting PEB for light harvesting and energy metabolism. The current work extends these studies in several notable ways. First, the authors show that HO and PcyX are usually associated with a virally encoded enzyme for synthesis of delta-aminolevulinic acid (ALA) via the Shemin or C4 pathway. This viral ALA synthase (vALAS) is shown to be functional both in extracts and via rescue of an E. coli strain lacking the equivalent enzymatic function. The associated HO and FDBR sequences are also shown to be functional, but the reaction product of the FDBR is PCB rather than PEB.

Overall, this work thus provides valuable insights. The presence of virally encoded enzymes for the committed step in tetrapyrrole biosynthesis extends the previous work from these groups and implicates an important role for tetrapyrrole biosynthesis in phage infection, even in the absence of photosynthesis by the host cell. Moreover, the demonstration of PcyA activity for this particular viral FDBR seems surprising based on the phylogenetic analysis presented in this work. This result thus provides new insight for the FDBR field and also may facilitate improved biosynthesis of bilins in heterologous systems, a growing field in its own right. The overall experimental quality is good, and the experimental methodology is very well described. To this reviewer, there are only a few points that could be considered in a revised version.

1. The apparent lack of any promoter induction could perhaps be described as “constitutive” rather than “leaky.” The demonstration of ALAS activity is thus quite important in interpreting the complementation result in E. coli. Have the authors considered drawing on the extensive library of known loss-of-function ALAS mutations? Expressing an inactive form of vALAS would be a very valuable addition to this line of experiments, because it should not rescue this strain no matter how leaky the promoter is.

In the functional complementation we used an auxotrophic E. coli strain ST18 which possesses the wild type lac regulon. The valas gene was expressed from a plasmid encoded copy under the control of the tac promoter. We believe that the basal level of expression from the tac promoter is sufficient for functional complementation. Therefore, we rephrased this paragraph accordingly.

Result section now reads (lines: 171 and following)

The plasmid encoded vAlaS under tac promoter control was able to rescue the ALA auxotrophic E. coli strain. The basal activity of the tac promoter was sufficient to enable functional complementation. Furthermore, we observed slightly slower growth of the complemented strain upon addition of inductor, probably due to increased metabolic burden overproducing the protein.

Based on the reviewer’s suggestion, a loss of function AlaS variant to further confirm the complementation results was constructed. We generated a non-functional vAlaS variant by site specific mutagenesis in the PLP cofactor binding site (vAlaS_K250A). The lysine residue K250 was exchanged by alanine. A plasmid encoding this vAlaS_K250A variant was used for functional complementation of the ALA auxotrophic E. coli ST18. The strain was not able to grow in absence of ALA. Thereby, we could confirm the functional complementation by the native vAlaS sequence. These data have been added to Fig. 5A and are now included in the revised version of the manuscript. The text in the results section was rewritten and changed accordingly (line 176 -179).

2. Can the light sensitivity of the porphyrin-producing strain be rescued by the co-expression of the viral HO, with or without the FDBR? This experiment could also provide insight into the potential value of such proteins during phage infection: such proteins could provide a means of rewiring the cell to produce more heme while avoiding potential phototoxicity.

Thank you for this suggestion. To answer this question, we performed coexpression experiments of valaS (of phage CB2) with ho1 and pebS (a different phage FDBR) in E. coli BL21 DE3. The strain expressing valaS showed accumulation of porphyrins, visible in culture color and fluorescence of cell lysate. The coexpression of valaS, with ho1 and pebS lead also to the accumulation of porphyrins in E. coli. The increased light-sensitivity due to porphyrin production could not be rescued by the co-expression of ho1 and pebS. Further we observed significant formation of phycoerythrobilin (PEB), which is actually expected, and not higher heme production.

cultures of BL21, BL21 + valaS, BL21 + valaS+ ho1+ pebS

Figure (for reviewer only): Cell lysates of *E. coli* strain BL21DE3 plasmid encoded vAlas (black) and vAlas + ho1 + pebS (red) were analysed for fluorescence emission (Ex. 400 nm): dashed lines.

The same samples were subsequently analysed in excitation scan for 615 nm fixed emission (solid line). Typical porphyrine spectra were observed for both lysates with absorption maximum at 400 nm and Q bands at 500 nm, 538 nm and 563 nm. Fluorescence emission was detected after excitation at 400 nm with a characteristic maximum at 615 nm.

In order to acknowledge this experiment, we added a sentence to the results section but as data not shown (now in line 195-196). Overall, we believe that the accumulation of porphyrins is rather an effect of our in vitro experiment in E. coli and will likely not occur in vivo during infection.

3. The authors provide convincing evidence for the presence of the first step in the Shemin (C4) pathway in some phages, encoded by vALAS and coupled with enzymes for production of bilins from heme (HO and PcyA/PcyX). They argue that increased flux through the heme biosynthesis pathway facilitates infection and that the Shemin pathway would be favorable relative to the Beale (C5) pathway because it requires a single enzyme to generate ALA and does not deplete charged Glu-tRNA molecules that would be useful for protein synthesis during infection. Have the authors looked for the presence of GluTR in phage genomes, with or without HO and FDBRs? This seems feasible to test, were GluTR to be in a cassette with FDBRs.

Thank you for the comment. We have not found any gltR and gsaM genes on the viral contigs that we have checked or where we have identified FDBR sequences.

4. The characterized FDBR seems quite close to known PcyX enzymes, despite its clear PcyA activity. Can the authors provide any insight into the mechanistic basis for this change in regiospecificity? Are there unusual active-site residues or some other explanation?

The characterized PcyA groups nicely with the previously characterized PcyA from Bradyrhizobium (Ledermann, 2016, Jaubert, 2007) which is a small subgroup directly between PcyA and PcyX. Characteristic for PcyX is Asp55 (which is Glu 76 in PcyA). We added a sentence in line 222 to clarify this.

5. The authors are effectively proposing that PCB can be supplied by phage-encoded PcyA to endogenous phytochrome photoreceptors, here in Pelagibacter spp. However, most bacterial phytochromes use BV as chromophore. Do the genomes of the proposed host cells encode candidate phytochromes that could incorporate a PCB precursor, as in cyanobacteria or Bradyrhizobium sp. ORS278?

Thank you for the comment. This is absolutely true, we just hypothesize that it could potentially be a phytochrome. As most hosts are unknown, we can only speculate whether a PCB binding phytochrome is present. However, to acknowledge the reviewers comment, we added a sentence that the host could also possess BBAGs (bilin biosynthesis associated globins) and cited Rockwell & Lagarias (2023)

Minor points:

1. Adding the Bradyrhizobium cassette to Fig. 2 would be useful for the reader, given that those sequences are highlighted as references in the trees. Also, what form of IMG accession is on this figure? It does not look like a scaffold code.

We added the Bradyrhizobium cassette to Figure 2 for clarity.

The accessions on the figure are according to the project of origin. GaXXXXXX are contigs retrieved from publicly available JGI Gold projects. SAMEA-named contigs were retrieved from a re-assembly of the Tara Oceans database used in Roitman et al., 2018. The "GROS-21" contigs originate from an unpublished database created by the authors that will be published soon, and is available upon request. Additionally, all contigs relevant to this paper can be found in their full sequence in Supplementary TableS2. A concise characterization of the phages, including name and database of origin can be found in Table S1

2. Adding peak wavelengths to Fig. 6B would be helpful.

The Figure was changed accordingly.

Reviewer #1 (Remarks to the Author):

The work entitled "Virus-encoded Shemin pathway highlights the importance of tetrapyrrole metabolism during host infection" by Wegner et al. discovered 5-ALA synthase genes of the Shemin pathway in bacteriophage genomes retrieved from both marine and freshwater environments. Viral alaSs (valaS) are predicted to be functional and are proved to be active in a 5-ALA-auxotrophic Escherichia coli strain. Methods description in this work are detailed enough. Data analyses and interpretation are appropriate and clear to support the conclusions. Together with the previous findings of viral tetrapyrrole-biosynthesis genes, the discovery of valaSs in this study provides new insights into the viral way of reprogramming host metabolism. This work is interesting and is important for furthering our understandings of the wide variety of AMGs.

Reviewer #2 (Remarks to the Author):

[No comments for authors]

Reviewer #3 (Remarks to the Author):

My review of the initial manuscript demonstrated its importance within the field. Overall, the revised version addresses my concerns and now seems ready for publication. However, I have one outstanding concern that is perhaps best dealt with at the galley stage: It seems there may be a typo in Fig. 2. It has been some time since this reviewer lived in the area, but should Fig. 2 really read "San Fransisco" and not "San Francisco" under the "Long AlaS" category?

Point-by point response to reviewer comments (author responses in italics):

Reviewer #1: Thank you

Reviewer #2: no comments from the reviewer

Reviewer #3: Thank you for pointing this out. Figure 2 has been changed accordingly.